# Report

# Metabolic effects of bezafibrate in mitochondrial disease

Hannah Steele[1,†], Aurora Gomez-Duran[2,3,†] (iD), Angela Pyle[1,4], Sila Hopton[4,5], Jane Newman[4,6], Renae J Stefanetti[6], Sarah J Charman[7], Jehill D Parikh[7], Langping He[4,5], Carlo Viscomi[3] (iD), Djordje G Jakovljevic[7], Kieren G Hollingsworth[7], Alan J Robinson[3], Robert W Taylor[4,5,6] (iD), Leonardo Bottolo[8,9,10,‡] (iD), Rita Horvath[2,‡] (iD) & Patrick F Chinnery[2,3,‡,*] (iD)

## Abstract

Mitochondrial disorders affect 1/5,000 and have no cure. Inducing mitochondrial biogenesis with bezafibrate improves mitochondrial function in animal models, but there are no comparable human studies. We performed an open-label observational experimental medicine study of six patients with mitochondrial myopathy caused by the m.3243A>G *MTTL1* mutation. Our primary aim was to determine the effects of bezafibrate on mitochondrial metabolism, whilst providing preliminary evidence of safety and efficacy using biomarkers. The participants received 600–1,200 mg bezafibrate daily for 12 weeks. There were no clinically significant adverse events, and liver function was not affected. We detected a reduction in the number of complex IV-immunodeficient muscle fibres and improved cardiac function. However, this was accompanied by an increase in serum biomarkers of mitochondrial disease, including fibroblast growth factor 21 (FGF-21), growth and differentiation factor 15 (GDF-15), plus dysregulation of fatty acid and amino acid metabolism. Thus, although potentially beneficial in short term, inducing mitochondrial biogenesis with bezafibrate altered the metabolomic signature of mitochondrial disease, raising concerns about long-term sequelae.

**Keywords** bezafibrate; metabolomics; mitochondrial disorder; mitochondrial DNA; mitochondrial encephalomyopathy
**Subject Categories** Pharmacology & Drug Discovery; Genetics, Gene Therapy & Genetic Disease; Metabolism

## Introduction

Mitochondrial diseases are genetically determined metabolic disorders of oxidative phosphorylation (OXPHOS) caused by mutations of mitochondrial DNA (mtDNA) or nuclear genes encoding mitochondrial proteins (Wallace, 2018). As a group, they affect ~ 1 in 5,000 (Gorman *et al*, 2015). The clinical features involve tissues with high-energy demands, especially striated muscle and the nervous system. Mitochondrial diseases cause substantial morbidity and have no cure.

Several new strategies are being developed as possible treatments for mitochondrial disorders. These include protein replacement; small molecules including antioxidants, amino acids and nucleotide supplementation; cell therapy; and gene manipulation (Nightingale *et al*, 2016). Although pre-clinical studies have seen the greatest progress, several are now under evaluation in clinical trials (Steele *et al*, 2017). One approach is to induce mitochondrial biogenesis, which is effective in animal models of mitochondrial diseases (Ahola-Erkkila *et al*, 2010; Viscomi *et al*, 2011; Cerutti *et al*, 2014; Peralta *et al*, 2016). Several compounds show promise in pre-clinical studies, including agonists of AMP-protein activated kinase (Viscomi *et al*, 2011; Peralta *et al*, 2016), peroxisome proliferator-activated receptor (PPAR) (Yatsuga & Suomalainen, 2012; Benit *et al*, 2017), sirtuins (Cerutti *et al*, 2014) and Nrf2 (Hayashi *et al*, 2017). Of these, bezafibrate has the most extensive pre-clinical evidence of efficacy in animal models (Dillon *et al*, 2012) and patient cell lines (Hofer *et al*, 2014). Bezafibrate has a favourable side-effect profile in humans but has been noted to cause liver toxicity in rodents, including mouse models of mitochondrial disease (Viscomi *et al*, 2011;

---

1 Institute of Genetic Medicine, Newcastle University, Newcastle upon Tyne, UK
2 Department of Clinical Neurosciences, University of Cambridge, Cambridge Biomedical Campus, Cambridge, UK
3 MRC Mitochondrial Biology Unit, University of Cambridge, Cambridge, UK
4 Wellcome Centre for Mitochondrial Research, Newcastle University, Newcastle upon Tyne, UK
5 NHS Highly Specialised Service for Rare Mitochondrial Disorders of Adults and Children, Newcastle upon Tyne Hospitals NHS Foundation Trust, Newcastle upon Tyne, UK
6 Institute of Neuroscience, Newcastle University, Newcastle upon Tyne, UK
7 Institute of Cellular Medicine, Newcastle University, Newcastle upon Tyne, UK
8 Department of Medical Genetics, University of Cambridge, Cambridge, UK
9 The Alan Turing Institute, London, UK
10 MRC Biostatistics Unit, University of Cambridge, Cambridge, UK
*Correspondence: Tel: +44 (0)1223 217091; E-mail: pfc25@cam.ac.uk
†These authors contributed equally to this work as first authors
‡These authors contributed equally to this work as last authors

Dillon *et al*, 2012; Yatsuga & Suomalainen, 2012). We, therefore, designed an open-label, investigator-led, non-randomised experimental medicine study to evaluate the effects of bezafibrate in patients with mitochondrial disease. We incorporated detailed molecular, biochemical and metabolomic profiling alongside clinical measures. Our primary aim was to evaluate the safety of inducing mitochondrial biogenesis with bezafibrate in patients with the m.3243A>G *MTTL1* mutation. Our secondary aim was to provide preliminary evidence of efficacy to power a subsequent randomised controlled trial.

# Results

## Participants and treatment

We studied six unrelated adults with mitochondrial myopathy due to the m.3243A>G *MTTL1* mutation in skeletal muscle (50–84% heteroplasmy, four female, mean age 50 years, range 44–57), identified from local clinics and the UK MRC Mitochondrial Disease Cohort. All had clinical evidence of proximal weakness on bedside testing (Medical Research Council grade 4+ or less). Four had diabetes and deafness, but no other clinical features of mitochondrial disease (Table 1). The entry and exclusion criteria are shown in Appendix Tables S1 and S2. The standard dose of bezafibrate used to treat dyslipidemia in the UK is 200 mg three times daily (TDS). Based on pre-clinical toxicology and efficacy studies in rodents, we calculated an equivalent dose range (Appendix Table S3) and treated with 200 mg TDS for 6 weeks, then 400 mg TDS in the absence of clinical toxicity. The study design is summarised in Fig 1. Needle muscle biopsies were performed with ethical approval and patient consent.

## Safety and tolerability

All six completed the study and treatment regime. There was no change in mean body mass index (BMI) during the treatment (Appendix Table S4). Although there was a trend towards a reduction in non-fasting triglyceride levels after 12 weeks of treatment, the mean level across all participants did not change significantly (Appendix Tables S5). One serious adverse event was reported: acute abdominal pain due to constipation which resolved overnight. This is a recognised complication of both bezafibrate treatment and mitochondrial disease. Four had mild hypoglycaemia leading to a reduction in the insulin dose in the three patients with insulin-treated diabetes. The other patient with mild hypoglycaemia had glucose intolerance under dietary control. The liver function tests remained normal. Three had a reversible reduction in estimated glomerular filtration rate (Fig EV1). FGF-21 and GDF-15 levels were elevated at baseline and increased further at 6 then 12 weeks with increased dose (Figs 2A and EV2) irrespective of gender, body mass index, body surface area or diabetes mellitus (Fig EV3). Unsupervised k-means clustering of the normalised serum metabolomics data separated the pre- and post-treatment groups (Group 1 vs. Groups 2 and 3; Fig EV4A), but there were no differences between weeks 6 and 12 (Fig EV4B). Investigation of the differentially regulated metabolites (Fig 2B, Dataset EV1) and

pathways (Fig 2C, Dataset EV2) highlighted changes in amino acid metabolism and tricarboxylic acid cycle intermediates (TCA cycle) between 0 and 6 and 12 weeks of treatment (Fig EV4C and D).

## Biochemistry and genetics

Targeted analysis of acyl-carnitines as a surrogate of β-oxidation showed a decrease in the levels of C3, C8, C14:1, C16:1, C18, C18:1 and C20 consistent with the induction of fatty acid oxidation on treatment (Figs 2D, and EV4E and F). At baseline, skeletal muscle complex I activity was low and complex III and IV activities were high or similar to control values relative to citrate synthase (Figs 2E and EV5A). Mean enzyme activities, including citrate synthase, did not change on treatment (Fig 2E), although there was considerable variation (Figs 2E, and EV5A and B). Quadruple immunofluorescence showed a reduction in the number of complex IV-deficient muscle fibres after treatment (*P*-value = 0.048, Fig 3A and B, Appendix Table S6, not detectable by Western blotting, Appendix Fig S1). Hierarchical cluster analysis and unsupervised *k*-means clustering of the skeletal muscle RNA-seq transcripts did not discriminate different groups before and after treatment (Fig 3C, Appendix Fig S2). Surprisingly, the analysis of differentially expressed transcripts (Dataset EV3) and Reactome Pathway Analysis (Dataset EV4) showed a significant decrease in genes involved in oxidative phosphorylation, including *MT-ND1*, *MT-ND2*, *MT-ND4*, *MT-ND-5*, and *MT-CYB* (Fig 3D and E). However, there was no change in the average mtDNA copy number in skeletal muscle with treatment (Appendix Fig S3). The mean percentage m.3243A>G mtDNA heteroplasmy in blood, urinary sediment and skeletal muscle did not change with treatment, although there was considerable variation in urinary sediment measurements (Fig 3F).

## Clinical measures

Mean submaximal exercise testing parameters did not change after treatment. Actigraphy showed no change in accelerometry, activity or sleep cycles with treatment. There was no difference in the skeletal muscle $^{31}$P-MRS phosphocreatine recovery time ($\tau_{1/2}$ PCr), adenosine diphosphate recovery time ($\tau_{1/2}$ ADP), or the ratio (ADP/PCr) with treatment (Appendix Fig S4). However, treatment was associated with an increase in end-systolic volume and end-systolic

**Table 1. Clinical characteristics of the six patients with the m.3243A>G *MTTL1* mutation and mitochondrial myopathy.**

| Subject | Sex | Age | Features in addition to myopathy | Symptom duration (years) | Baseline NMDAS |
|---|---|---|---|---|---|
| P1 | F | 52 | MIDD | 17 | 36 |
| P2 | F | 52 | | 9 | 18 |
| P3 | M | 44 | MIDD | 28 | 33 |
| P4 | M | 49 | | 11 | 19 |
| P5 | F | 57 | MIDD | 18 | 21 |
| P6 | F | 46 | MIDD | 11 | 19 |

MIDD, mitochondrially inherited diabetes and deafness; NMDAS, Newcastle Mitochondrial Disease Adult Scale.

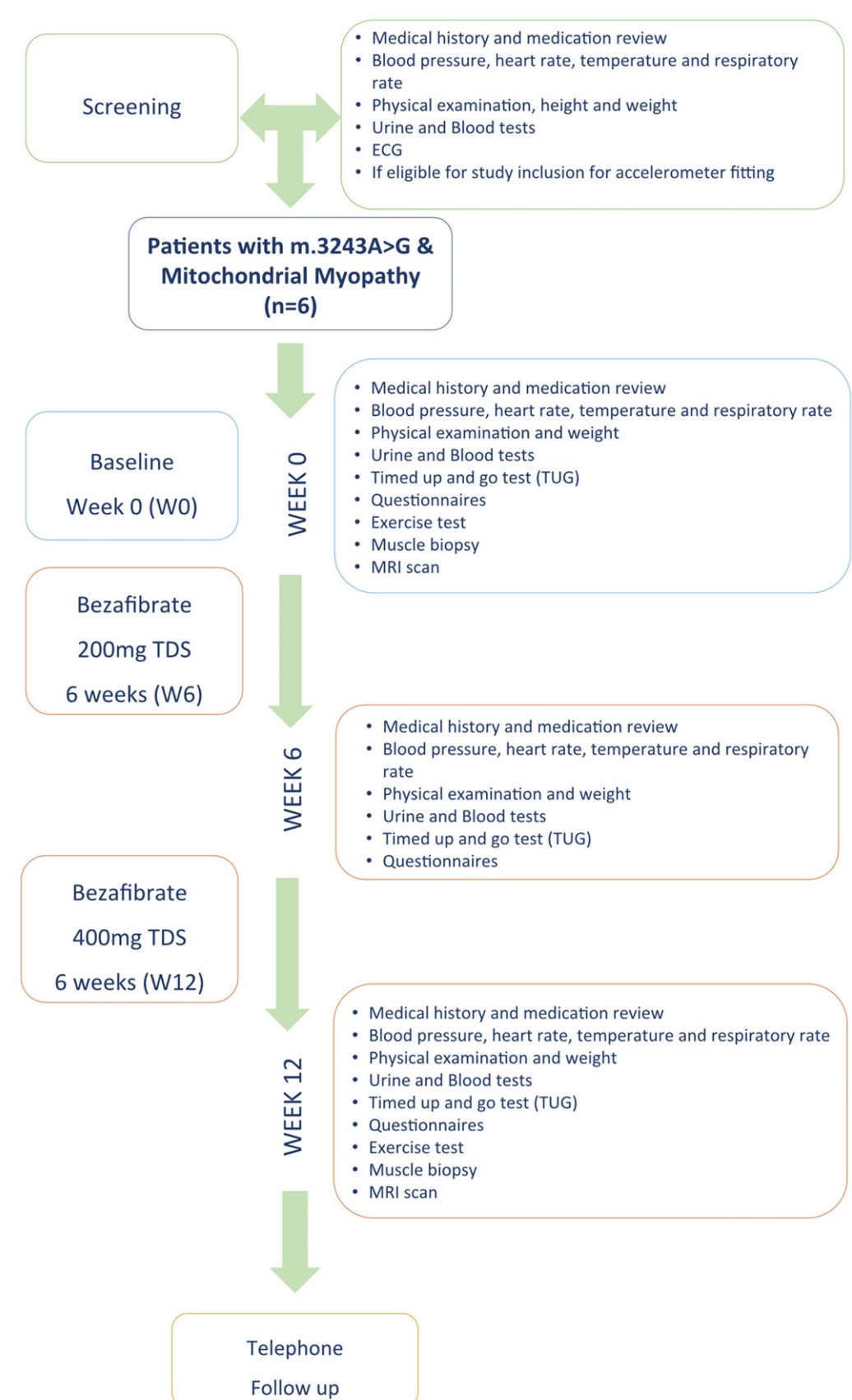

**Figure 1.  Summary of the study design**.

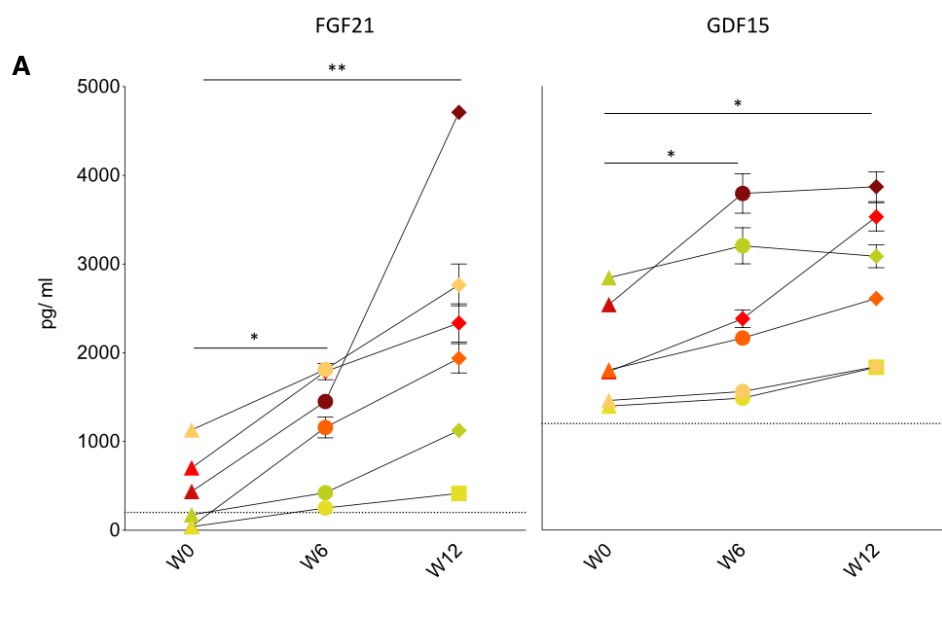

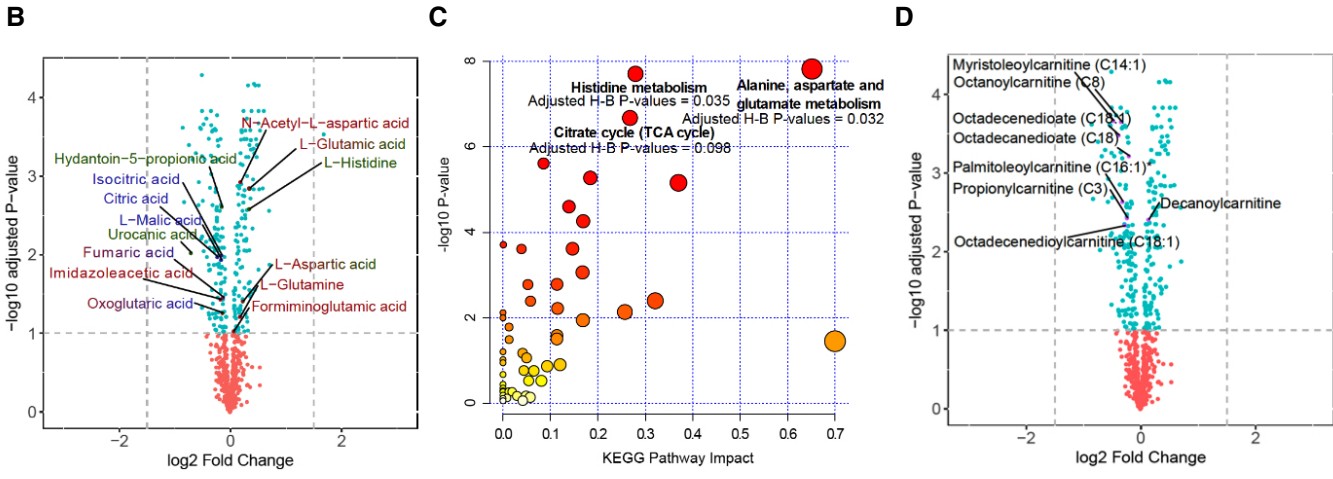

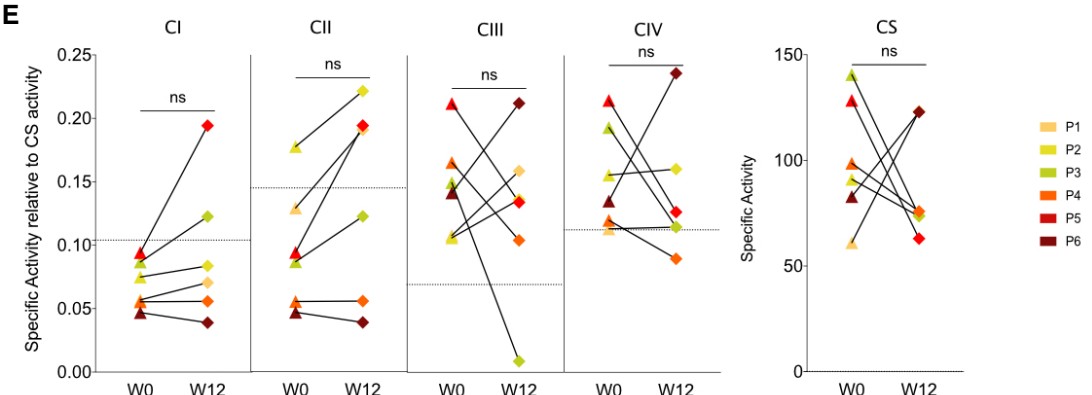

Figure 2.

**Figure 2. Metabolic effects of bezafibrate in patients with the m.3243A>G *MTTL1* mutation.**

W0, W6 and W12 refer to the week of study.

A  Serum FGF-21 and GDF-15 levels before, during and after treatment (two-sided Wilcoxon signed-rank test, *P*-value = 0.031 in all significant cases) (see Fig EV2 for details where P1–P6 refer to the individual patients).

B  Volcano plot showing differences in metabolite levels between 0 and 6 and 12 weeks of treatment. Metabolites in histidine, alanine, aspartate and glutamine metabolism, and the TCA cycle pathways are annotated at 10% false discovery rate (see Dataset EV1 for the whole list of differentially regulated metabolites. Although some other metabolites showed a more pronounced difference, these did not fit into a recognised pathway). Metabolite labels are colour-coded according to the identified KEGG pathways (dark red = "Alanine, aspartate and glutamate metabolism", dark green = "Histidine metabolism" and dark blue = "Citrate cycle (TCA cycle)", with transition colour code if the metabolite belongs to more than one KEGG pathways).

C  Scatterplot depicting *P*-values from the metabolites Pathway Enrichment Analysis (y-axis) and impact values from Pathway Topology Analysis (x-axis) (see Dataset EV2 for details). KEGG pathways at 10% adjusted Holm–Bonferroni *P*-value are highlighted.

D  Volcano plot of acyl-carnitine metabolites between 0 and 6 and 12 weeks of treatment with highlighted differential metabolites at 10% FDR.

E  Respiratory chain complex and citrate synthase activity in skeletal muscle before and after 12 weeks of treatment. No significant differences were detected before and after 12 weeks of treatment by using the two-sided Wilcoxon signed-rank test with an empirical level of significance.

Data Information: In (A) and (E), horizontal dotted lines denote the mean values in healthy age-matched controls. In (E), solid horizontal lines and bars represent the mean ± SD of the technical replicates at each time point. Horizontal lines on (A) and (E) refer to statistical comparisons of the mean values, where ns = non-significant, *\*P*-value ≤ 0.05.

index (Fig 3G). The mean questionnaire scores did not change following treatment, including the NMDAS and muscle-specific scores (Appendix Table S7). There was no correlation between the level of m.3243A>G heteroplasmy or clinical features and the clinical or biochemical responses to treatment.

## Discussion

As expected, we detected evidence of on-target effects of bezafibrate through the induction of fatty acid β-oxidation (Bonnefont *et al*, 2010) and a trend towards decreased triglyceride levels. However, even using doses higher than typically used to treat dyslipidaemia, bezafibrate had a minimal impact on markers of mitochondrial biogenesis. Similar to other studies using PPAR agonists, we observed dose-dependent increases in serum GDF-15 and FGF-21 (Inagaki *et al*, 2007; Yatsuga & Suomalainen, 2012), which have been proposed as biomarkers of mitochondrial disease (Lehtonen *et al*, 2016). This raises the possibility that treatment led to an exacerbation of the mitochondrial pathology in patients with the m.3243A>G mutation, although given the lack of specificity of GDF-15 and FGF-21, it is also plausible that the changes we observed were due to the direct effects of bezafibrate on metabolism independent of oxidative phosphorylation.

Following bezafibrate treatment, we also saw an alteration of metabolic pathways involving glutamine metabolism and the tricarboxylic acid (TCA) cycle (Buzkova *et al*, 2018), as well as an increase in the level of several amino acids including histidine, alanine, aspartate and N-acetyl-aspartate (NAA) (Thompson Legault *et al*, 2015). Glutamine is an abundant circulating amino acid and has an anaplerotic role, replenishing TCA cycle intermediates, which drive the mitochondrial respiratory chain to generate reducing equivalents (Mullen *et al*, 2014; Chen *et al*, 2018). In the context of mtDNA mutations, glutamine is reduced by the TCA cycle leading to the synthesis of aspartate, which is essential for nucleotide synthesis (Chen *et al*, 2018). Thus, increased amino acid levels, including glutamate, could be part of a compensatory response to the primary OXPHOS defect (Buzkova *et al*, 2018; Chen *et al*, 2018). Similar findings have been observed for aspartate, with effects on reductive carboxylation (Chen *et al*, 2018; Sullivan *et al*, 2018). Likewise, increased histidine has been

observed in models of mitochondrial disease (Vo *et al*, 2007; Chen *et al*, 2018), where it can act as a source of folate (Mao *et al*, 2004) and TCA cycle pre-intermediates such as glutamate, which are then degraded to a-ketoglutarate (Alifano *et al*, 1996). It is therefore possible that bezafibrate provokes or enhances an endogenous compensatory response to the primary metabolic defect that resembles a fasting response (Crooks *et al*, 2014; Tan *et al*, 2017), despite the patients being well-nourished throughout, and with no associated weight loss. However, given the observed increase in all three amino acids, it remains puzzling that, overall, the TCA metabolites decreased on treatment. The reasons for this are not clear, but could reflect the effects of increased aspartate levels, which are known to inhibit alpha-ketoglutarate utilisation (Chen *et al*, 2018), with knock-on effects on other TCA cycle intermediates.

Our study was not designed to detect significant changes in clinically meaningful parameters, but the data will provide a useful basis for future power calculations. It is encouraging that we observed an improvement in parameters of cardiac function on MRI that are known to be impaired in patients with m.3243A>G (Hollingsworth *et al*, 2012). The clinical significance of these findings is unclear at present, but they support the use of cardiac MRI as a biomarker to monitor treatment in mitochondrial diseases (Steele *et al*, 2017), which is sensitive to changes over a short period and correlates with the underlying biochemical defect in skeletal muscle (Hollingsworth *et al*, 2012).

Importantly, we observed considerable variability in several proposed biomarkers, including exercise physiology and m.3243A>G heteroplasmy in urinary sediments, explaining why these do not always correlate with clinical severity. On the other hand, being sensitive to changes at the single-cell level, quadruple immunofluorescence shows promise when compared bulk measures of respiratory chain protein abundance and function in skeletal muscle. We did not observe a similar change in either the respiratory chain activity or protein abundance on Western blots, probably because the changes only affected a small proportion of the individual muscle fibres, and were thus not detectable in an analysis of a muscle homogenate. Thus, although the quadruple immunofluorescence may be a sensitive biomarker indicative of a treatment response, the clinical significance of these findings remains to be determined.

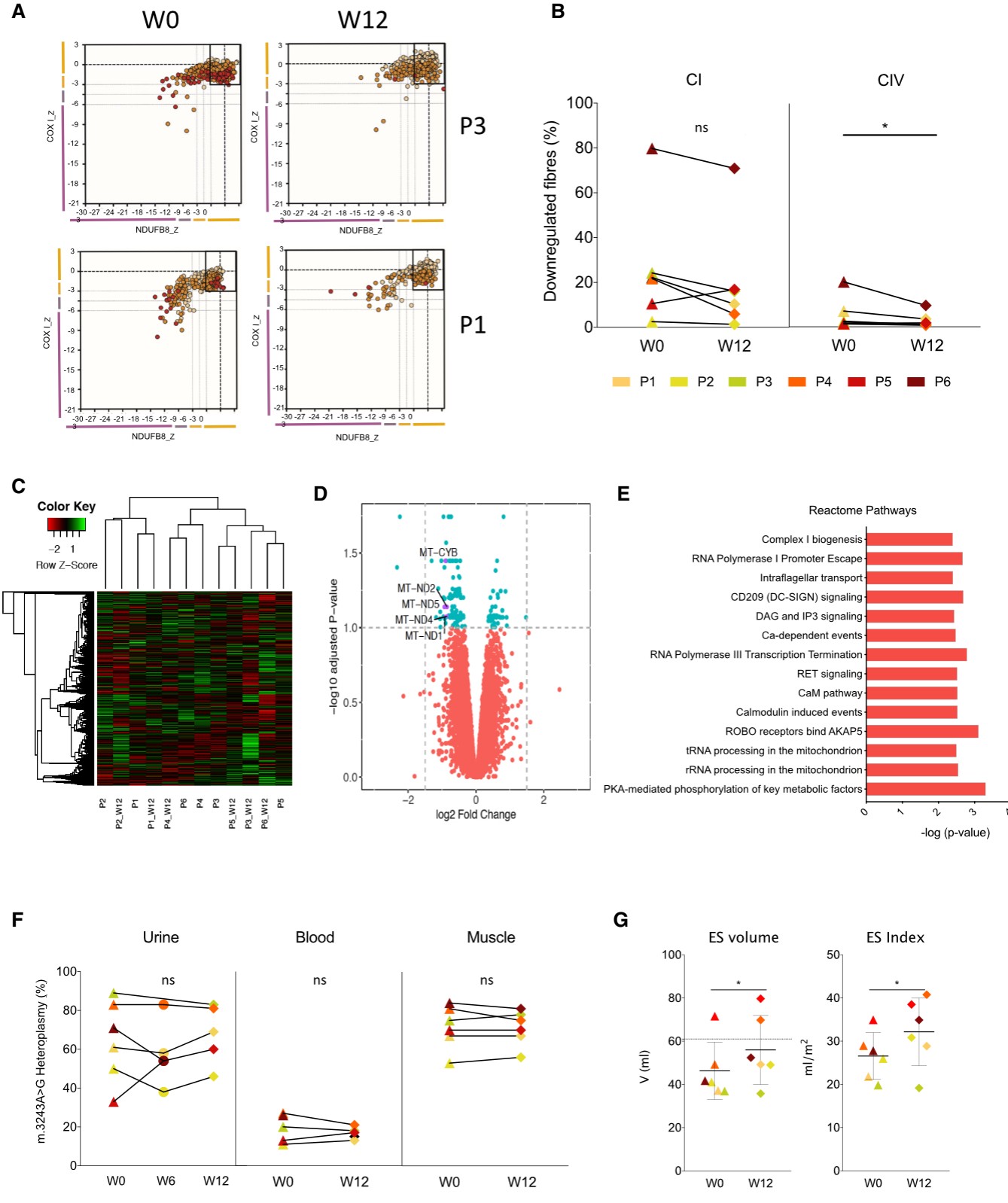

**Figure 3.**

**Figure 3.  Mitochondrial biogenesis and clinical effects of bezafibrate in patients with the m.3243A>G *MTTL1* mutation.**

P1–P6 refer to individual patients.

A   Representative examples of the quadruple immunofluorescence quantification of skeletal muscle fibres before and after treatment. The box in the top right-hand corner indicates the normal range, using two antibodies to NDUFB8 and COX I.

B   Quantitative quadruple immunofluorescence of skeletal muscle fibres in the six patients showing a decrease (two-sided Wilcoxon signed-rank test with an empirical level of significance) in the proportion of complex IV-deficient skeletal muscle fibres before and after 12 weeks of treatment (empirical *P*-value = 0.048).

C   Hierarchical cluster analysis of skeletal muscle RNA-seq transcripts before and after treatment showing no clear separation between untreated and treated patients.

D   Volcano plot depicting differentially expressed RNA-seq genes at 10% FDR after 12 weeks of treatment. Significantly different mammalian mitochondrial genes (MitoCarta 2.0) are annotated.

E   Reactome Pathway Analysis of the differentially expressed genes at 10% FDR after 12 weeks of treatment (see Expanded View document for details).

F   Percentage level m.3243A>G in blood, urinary sediment and skeletal muscle. No differences were detected by using the two-sided Wilcoxon signed-rank test with an empirical level of significance. W = week of the study.

G   Cardiac parameters before and after treatment (two-sided Wilcoxon signed-rank test with an empirical level of significance). ES = end-systolic volume (ml) (empirical *P*-value = 0.047) and index (ml/m$^2$) (empirical *P*-value = 0.046), measured by MRI. Horizontal lines on (B), (F) and (G) refer to statistical comparisons of the mean values, where ns = non-significant, \**P*-value ≤ 0.05. For normal reference ranges, see Hollingsworth *et al* (2012).

---

It should be noted that our results are based on observations in a specific subgroup of patients with mitochondrial disease due to the m.3243A>G mutation. However, they do suggest that bezafibrate should be used with caution to treat dyslipidaemia in patients with mitochondrial disorders. Future studies inducing mitochondrial biogenesis through PPAR activation should proceed with caution, with careful monitoring for possible toxicity. Measuring blood metabolomic biomarkers, FGF-21, and GDF-15 in addition to serum creatine kinase would provide some reassurance if levels do not change on treatment.

## Materials and Methods

In overview, the following data were collected: (i) Measures of safety and tolerability at 0, 6 and 12 weeks including adverse events, haematological and biochemical blood tests, creatine kinase; FGF-21, GDF-15 and serum metabolomics; (ii) respiratory chain enzyme activity in skeletal muscle at baseline and 12 weeks; (iii) percentage of cytochrome *c* oxidase (COX) muscle fibres by histochemistry, quantification of respiratory chain complexes I and IV proteins NDUFB8 and COXI by quadruple fluorescence immunohistochemistry (Rocha *et al*, 2015) and Western blotting, all at 0 and 12 weeks; (iv) muscle RNA-seq at 0 and 12 weeks; (v) percentage heteroplasmy of the m.3243A>G in blood and urinary sediment at 0, 6 and 12 weeks, and muscle at 0 and 12 weeks; total mtDNA copy number in skeletal muscle at 0 and 12 weeks; (vi) submaximal exercise testing, $^{31}$P-magnetic resonance spectroscopy ($^{31}$P-MRS) in skeletal muscle, cardiac function by MRI, accelerometry and activity, timed up and go (TUG), and the following questionnaires: International Physical Activity Questionnaire (IPAQ), Newcastle Mitochondrial Disease Adults Scale (NMDAS), the Newcastle Mitochondrial Disease Quality of Life Score (NMQ), the Fatigue Impact Score (FIS), all at 0 and 12 weeks. See the Expanded View document for detail methods and citations.

### Detailed methods

#### Serum FGF-21 and GDF-15

Blood samples were collected in a serum-separating tube at times 0, 6 and 12 weeks of the study. Samples were centrifuged at 5,000 *g* for 10 min, and the resulting serum was snap frozen. FGF-21 and

GDF-15 were measured by ELISA from BioVendor and R&D systems, respectively, following manufacturer's instructions.

#### Serum metabolomics analysis

Sample preparation, QA/QC and Ultrahigh Performance Liquid Chromatography-Tandem Mass Spectroscopy (UPLC-MS/MS) were performed at Metabolon© as well as compound identification, quantification and data curation as described previously (Shin *et al*, 2014). A total of 867 different metabolites were measured on human serum from six patients over three-time points at 0 (untreated), 6 and 12 weeks (treated). Among the measured metabolites, only four were partially characterised (untargeted), whilst known (targeted) metabolites spanned a wide range of biochemical classed ("super-pathways") including 187 "Amino Acid" (22%), 27 "Carbohydrate" (3%), 28 "Cofactors and Vitamins" (3%), 11 "Energy" (1%), 409 "Lipid" (47%), 37 "Nucleotide" (4%), 37 "Peptide" (4%) and 127 "Xenobiotics" (15%). Normalisation and imputation of serum metabolites were performed as previously described (Shin *et al*, 2014; Krumsiek *et al*, 2015). Briefly, data normalisation (N) consisted of the following steps: (N.1) partially characterised metabolites and metabolites belonging to "Xenobiotics" biochemical class were removed from the analysis, reducing the number of targeted metabolites to 736; (N.2) each metabolite raw value was rescaled to have median 1 to adjust for variation due to instrument run-day tuning differences; (N.3) a log transformation with base 10 was applied to all the metabolites; (N.4) after transformation, data points lying more than four standard deviations from the mean of each metabolite concentration were excluded. For the imputation (I) of missing values, we employed the KNN-TN method (Shah *et al*, 2017), which consists of the following steps: (I.1) Estimation of the detection level (DL) of the machine to be the minimum observed value for the whole dataset; (I.2) maximum likelihood estimation (MLE) of $\mu_m$ and $\sigma_m$, assuming that each metabolite $m$ ($m = 1,\ldots,736$) follows a left-truncated (on the DL) Gaussian distribution with mean $\mu_m$ and standard deviation $\sigma_m$; (I.3) standardisation of each metabolite using the MLEs of $\mu_m$ and $\sigma_m$; (I.4) for each metabolite $m$ with a missing value in sample $i$, detection of its $K = 10$ closest metabolites (which have an observed value for their $i^{th}$ sample) using the *k*-nearest neighbours (KNN) algorithm; (I.5) imputation of the missing value with a weighted average of the $K$ values found in (I.4). The weights are functions of the Pearson correlations between the metabolite with missing values and its $K$

closest metabolites; (I.6) transformation of each metabolite back to the original scale as it was before step (I.3).

Differential analysis of 736 targeted serum metabolites was performed by using Limma (Ritchie *et al*, 2015) after correcting for treatment time points (0, 6 and 12 weeks) using a linear mixed model with treatment time points as a random effect. For each contrast (6- and 12-week treatment vs. untreated, 6-week treatment vs. untreated, 12-week treatment vs. untreated, and 12-week treatment vs. 6-week treatment), significant differential metabolites were declared at 10% FDR (Benjamini *et al*, 2001). 267 (36%) significant differential metabolites were discovered at 10% FDR in the 6- and 12-week treatment vs. untreated contrast. Pathway Enrichment Analysis was performed by using MetaboAnalyst 4.0 (Chong *et al*, 2018) with 176 (23%) significant differential metabolites mapped in KEGG pathways. Given the lack of pathways annotation for a large fraction of differential metabolites, significantly enriched pathways were declared at a conservative 10% Holm–Bonferroni correction. Finally, Pathway Topology Analysis was performed by selecting the relative-betweenness centrality measure (ranging between 0 and 1), which quantifies the importance of a subgroup of metabolites in a given metabolic pathway. Residuals from the linear mixed model were also used to perform the $k$-means unsupervised clustering of the participants' treatment time points.

### Respiratory chain enzyme activity and citrate synthase activity

The activities of mitochondrial respiratory chain enzymes and the matrix marker citrate synthase were determined spectrophotometrically in skeletal muscle homogenates as described, within a UKAS-accredited (ISO 15189) laboratory (Kirby *et al*, 2007).

### Immunoblot analysis

Approximately 25–45 mg of snap frozen muscle was lysed in RIPA buffer (10×) (Sigma) buffer containing 1× cOmplete™ protease inhibitor (Roche). Samples were homogenised using a Precellys®24 mechanical tissue homogeniser and centrifuged at 14,000 $g$ for 10 min at 4°C. The supernatant, containing the soluble protein, was kept. Protein concentration in each sample was determined using the Pierce™ BCA Protein Assay Kit (Thermo Fisher) using manufacturer's instructions. In each electrophoresis, 20 µg of each sample was loaded NuPAGE® Bis-Tris Precast Midi Protein Gels with MES (Invitrogen®) following the manufacturer's conditions. SeeBlue® Plus2 Pre-stained Protein Standard from Invitrogen® was used in each electrophoresis as protein size marker. The separated proteins were transferred to polyvinylidene fluoride membranes using the iBLOT system (Invitrogen®). The resulting blots were probed overnight at 4°C with primary antibodies with the appropriate concentration following manufacture's condition with small adaptations. Briefly, OXPHOS Human WB Antibody Cocktail Complex Ab110411 (1:250) and Porin/VDAC1 Ab434 (1:1,000). After the primary antibody, blots were incubated for 1 h with secondary antibodies [IR Dye® 680 Goat anti-Rabbit Li-Cor® #A32729 (1:2,000) and IR Dye® 680RD Goat Anti Mouse Li-Cor® 926-68070 (1:1,000)] and the membrane was imaged using the Li-Cor® Odyssey® CLx Imager. The bands for each antibody were quantified, aligned and cropped using bands that were quantified using the Li-Cor® Odyssey® CLx Imager inbuilt analysis software, and the OD was used as a value for statistical purposes. OD values corrected by porin in each case.

### Muscle histochemistry and immunocytochemistry

Sequential COX/SDH histochemistry (COX/SDH) and quantitative, quadruple immunofluorescence were carried out according to established, clinically validated protocols within a UKAS-accredited (ISO 15189) laboratory (Old & Johnson, 1989; Rocha *et al*, 2015).

### RNA-seq analysis

Total RNA was extracted from muscle samples using RNeasy Mini Kit as per manufacturer's instructions (Quiagen, UK). RNA integrity and quantity were assessed by RNA ScreenTape Analysis (Agilent). Samples with RIN number higher than 8.5 were used for RNA-seq. RNA-seq was performed with TruSeq Stranded Total RNA with RiboZero following manufacture's conditions. Counts from the raw RNA-seq bezafibrate FastQ files were generated via Salmon (Patro *et al*, 2017), which also maps the reads to the reference transcriptome. Annotation of the genes was obtained by using the R package biomaRt (Durinck *et al*, 2009), whilst the list of mammalian mitochondrial genes was downloaded from MitoCarta2.0 (Calvo *et al*, 2016). Normalisation and filtering of raw count data were performed by using the R package edgeR (Robinson *et al*, 2010). Since the ratio of the largest library size to the smallest is less than 3-fold (1.31), mean–variance relationship is modelled with an empirical Bayes prior trend with read counts converted to $\log_2$-counts-per-million (log CPM). Log CPM gene expression measurements were used to perform the k-means unsupervised clustering of the participants' treatment time points.

Differential analysis of log CPM gene expression (see Table E5) was performed by using Limma (Ritchie *et al*, 2015) with paired blocking (12-week treatment vs. untreated). Significant differential genes were declared at 10% FDR (Benjamini & Hochberg, 1995). Pathway analysis was performed by Reactome (Fabregat *et al*, 2018).

### Molecular genetics

Total DNA was extracted from skeletal muscle (tibialis anterior), urinary sediment, and blood using DNeasy Blood & Tissue Kit following manufacturer's instructions (Quiagen, UK). Measurement of m.3243A>G heteroplasmy (de Laat *et al*, 2016) and mtDNA levels (Pyle *et al*, 2015) were quantified following previously reported methods.

### Cardiac MRI

Participants were imaged supine using a 6-channel cardiac coil and ECG gating. Long axis 4 chamber views and a short axis stack with balanced steady-state free precession images obtained during breath-holding and covering the entire left ventricle (field of view (FOV) $350 \times 350$ $mm^2$, repetition time/echo time (TR/TE) = 3.7/1.9 ms, turbo factor 17, flip angle (FA) 40°, slice thickness 8 mm, 25 phases, resolution 1.37 mm) were acquired. Image analysis was undertaken using the cardiac analysis package of the ViewForum workstation (Philips, Best, NL). Manual tracing of the epicardial and endocardial borders was performed on the short axis slices at end-systole and end-diastole. Left ventricular mass, ejection fraction, end-systolic and end-diastolic parameters were calculated. The eccentricity ratio was calculated as the ratio of LV mass to end-diastolic volume, with increased values demonstrating the presence of concentric remodelling. To

standardise measurements for participant size, body surface area was calculated using subjects' weight and height (Du Bois & Du Bois, 1989).

### Clinical rating scales

International Physical Activity Questionnaire (IPAQ), Newcastle Mitochondrial Disease Adults Scale (NMDAS), the Newcastle Mitochondrial Disease Quality of Life Score (NMQ), the Fatigue Impact Score (FIS) (Schaefer et al, 2006).

### Cycle ergometry

All individuals followed the same exercise protocol using an electronically braked cycle ergometer (Corival Lode BV, Groningen, The Netherlands) and gas analysis system (Metalyzer 3B; Cortex, Leipzig, Germany). Cardiac output was measured using validated bioreactance techniques (NICOM, Cheetah Medical, Delaware, USA) using two electrodes placed over the trapezius muscle on either side of the torso; and two placed on the lower posterior torso lateral to the margin of the latissimus dorsi muscle (Jones et al, 2015). Prior to exercise testing, participants underwent a 12-lead electrocardiogram (ECG; Custo med GmbH, Ottobrunn, Germany) and blood pressure monitoring (Suntech Tango+, Suntech Medical Ltd, Oxford, UK) in a semi-recumbent seated position. Expired gases and cardiac output were collected for at least 5 min at rest. During exercise testing, the ECG and cardiac output were monitored continuously and blood pressure was measured every 2 min. Both ECG and blood pressure (BP) were monitored for at least 5 min after the exercise test had been terminated in order to monitor participant recovery. Participants undertook a maximal graded cardiopulmonary exercise stress test using an incremental protocol of 10 W/min. Participants were requested to maintain a cadence of 60 rpm on the ergometer. In order to avoid influencing the results of repeat exercise testing at week 12, the baseline test results were not available to the investigators for review until after the second test. The exercise test was terminated by the presence of severe dyspnoea; muscle fatigue; physical exhaustion (defined as a respiratory exchange ratio > 1.1; or Borg RPE (Rating of Perceived Exertion) score > 18; or failure to increase oxygen consumption despite increasing exercise intensity) or where participants were unable to maintain requested cadence despite encouragement. Valid tests were those terminated due to physical exhaustion as defined above. Peak oxygen consumption and peak work rate were defined as average values over the last 30 s of the exercise (before termination of the test). Cardiac output (CO) was calculated as the product of stroke volume (SV) and heart rate (HR), and the arterial-venous difference (a-VO$_2$ diff) was calculated as the ratio between oxygen consumption and cardiac output.

### Accelerometry

Physical activity was measured using a tri-axial accelerometer (GENEActiv, Activinsights Ltd, UK) worn at the wrist for seven consecutive days on three occasions over the study: (i) between screening and baseline visits; (ii) between weeks five and six; (iii) and between weeks 11 and 12. Participants were instructed to wear the devices at all times during the 7-day monitoring period including whilst sleeping, bathing or showering and requested that the device be worn on the same wrist at each of the three measurement times. The sampling frequency was set to 50 Hz. Raw accelerometry data were run through R-package GGIR version 1.2.8 (www.cran.r-project.org). The first and last hour of the measurement was excluded as they are expected to be influenced by the monitor distribution and collection procedure (van Hees et al, 2013, 2015). Criteria for valid wear were pre-defined as a minimum of 16 h of data per day of wear, with at least 5 of the 7 days meeting these criteria, and at least one of these days being a weekend. Any monitor non-wear detected was imputed using available data from a similar time-period (e.g. day or night) and type of day (weekend or weekday). Captured data enabled assessment of: acceleration (milligravity, where 1 mg = 0.001 g = $0.001 \times 9.8$ m/s$^2$ = 0.001 × gravitational acceleration); activity duration (time) spent in different intensities (sedentary, light, moderate, vigorous); sleep duration, sleep efficiency and wake time overnight. Light, moderate and vigorous activity was calculated using $\geq 40$, $\geq 100$ and $\geq 120$ m$g$ cut-offs, respectively (Charman et al, 2016).

### Timed up and go

Participants were assessed at weeks 0, 6 and 12. Participants were instructed to sit in a standard chair kept in MoveLab and provided with standardised instructions: "When I say 'go' I want you to stand up and walk to the line, turn, walk back to the chair and sit down again. Walk at your normal pace". Subjects were timed from being told to "go" until their return to a seated position. The test was repeated three times at each assessment. Statistical analysis was undertaken using IBM SPSS, version 23 (SPSS Inc., Chicago, IL, USA).

## Statistical and bioinformatic analysis

Statistical testing was performed by using the Wilcoxon signed-rank test (paired data), the Mann–Whitney test, both with an empirical level of significance (Hollander & Wolfe, 1973), and the Kruskal–Wallis test followed by the Tukey honest significant differences test on the ranked data if the null hypothesis of Kruskal–Wallis test was rejected (Yandell, 1997). All statistical tests are two-sided. Permutation test for the Mann–Whitney test was conducted using the R package coin (Hothorn et al, 2008). For the Wilcoxon signed-rank test (paired data), we implemented an in-house procedure which flips at random for each patient the observations at different time points preserving the paired nature of the data. Given the small number of combinations (6! = 720), we calculated the exact empirical null distribution. The empirical P-value was obtained by counting the number of times the Wilcoxon signed-rank test statistic based on the observed data was greater than the same test statistic on the permuted data.

## Approvals, sponsorship and consent

The study was registered in advance on ClinicalTrials.gov (NCT02398201) and EudraCT (2015-000508-24), with Newcastle upon Tyne Hospitals NHS Trust as sponsor. Ethical approval was provided by Newcastle and North Tyneside 1 National Research Ethics Committee (15/NE/0166, IRAS 167358). The full protocol is available with the full text of this article at https://clinicaltrials.gov/ct2/show/NCT02398201. Informed consent was obtained from all

**The paper explained**

**Problem**

Mitochondrial disorders affect 1/5,000 but have no cure. Inducing mitochondrial biogenesis with bezafibrate improves mitochondrial function in animal models, but there are no comparable human studies. Abnormal liver function has been observed in pre-clinical rodent studies, but it is not known whether humans are similarly vulnerable to hepatic side effects. To address this, we performed an open-label observational study of six patients with mitochondrial myopathy caused by the m.3243A>G *MTTL1* mutation. Participants received 600–1,200 mg bezafibrate for 12 weeks. Our aim was to determine the effects of bezafibrate on mitochondrial metabolism, whilst providing preliminary evidence of safety and efficacy using biomarkers.

**Results**

There were no clinically significant adverse events. Liver function was not affected by the treatment. Clinical and biochemical markers showed marked inter-individual variability. We saw a reduction in the number of complex IV-immunodeficient muscle fibres and improved cardiac function. However, this was accompanied by an increase in serum disease biomarkers of mitochondrial disease, including fibroblast growth factor 21 (FGF-21), growth and differentiation factor 15 (GDF-15) and amino acids including glutamine and aspartate.

**Impact**

Although potentially beneficial in short term, inducing mitochondrial biogenesis with bezafibrate affects the metabolomic signature of mitochondrial disease, raising concerns about long-term sequelae. These results can be used to design a definitive randomised controlled trial for drugs increasing mitochondrial biogenesis, but long-term monitoring will be required to minimise the risk of adverse effects. At present, there is no compelling evidence to use bezafibrate as a treatment for mitochondrial disease due to the m.3243A>G mutation, and if prescribed for dyslipidaemia it should be used with caution.

participants, and the experiments conformed to the principles set out in the WMA Declaration of Helsinki and the Department of Health and Human Services Belmont Report.

## Data availability

The datasets produced in this study are available in the following databases: RNA-Seq data: SRA accession number PRJNA608421.

**Expanded View** for this article is available online.

## Acknowledgements

We are grateful to Michael Hanna (UCL Institute of Neurology) and Grainne Gorman (Newcastle University) for their role in the trial oversight committees, and for the assistance of Andrew Blamire, Alex Bright, Jennifer Duff, Gavin Falkous, Lesley Hall, Cecilia Jimenez-Moreno, Robert McFarland, Yi Ng, Robert Pitceathly, Alan Robinson, Michael Trenell and Patrick Yu-Wai-Man. We would like to acknowledge the Wellcome Centre for Mitochondrial Research Patient Cohort: A Natural History Study and Patient Registry (Research Ethics Committee Reference 13/NE/0326) for facilitating recruitment. PFC is a Wellcome Trust Principal Research Fellow (212219/Z/18/Z), and a UK NIHR Senior Investigator, who receives support from the Medical Research Council Mitochondrial Biology Unit (MC_UU_00015/9), the Evelyn Trust and the National Institute for Health Research (NIHR)

Biomedical Research Centre based at Cambridge University Hospitals NHS Foundation Trust and the University of Cambridge. RWT is supported by the Wellcome Centre for Mitochondrial Research (203105/Z/16/Z), the Medical Research Council (MRC) International Centre for Genomic Medicine in Neuromuscular Disease, the Mitochondrial Disease Patient Cohort (UK) (G0800674), the Lily Foundation, the UK National Institute for Health Research Biomedical Research Centre in Age and Age-Related Diseases award to the Newcastle upon Tyne Hospitals NHS Foundation Trust, the MRC/EPSRC Molecular Pathology Node and the UK NHS Highly Specialised Service for Rare Mitochondrial Disorders of Adults and Children. RH is a Wellcome Senior Investigator (109915/Z/15/Z) and supported by the Medical Research Council (UK) (MR/N025431/1), the Newton Fund (UK/Turkey, MR/N027302/1), the European Research Council (309548) and the Wellcome Trust Pathfinder Scheme (201064/Z/16/Z). LB is supported by The Alan Turing Institute under the Engineering and Physical Sciences Research Council grant EP/N510129/1. The views expressed are those of the author(s) and not necessarily those of the NHS, the NIHR or the Department of Health.

## Author contributions

PFC and RH conceived the study and sought funding. The protocol was designed by HS, RH and PFC. HS led the study and performed the clinical, laboratory and MRI measures with AP, JN, RJS, SJC, and JDP. LB, AG-D and AJR analysed the metabolomics and transcriptomic data, and LB supervised the statistical and bioinformatics analysis. RWT supervised the biochemical, histochemical, immunofluorescence and mtDNA analyses. DGJ supervised the exercise testing. LH performed the biochemical measurements of respiratory chain enzyme activities. SH performed the quadruple immuno-histochemical assessment of OXPHOS protein levels. KGH supervised MRI data acquisition and analysis. PFC, AG-D, HS, RH, CV and LB wrote the paper with input from all authors, who approved the final version of the manuscript.

## Conflict of interest

The authors declare that they have no conflict of interest.

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
