## [Review Process File · EMBO Molecular Medicine]

Metabolic effects of bezafibrate in mitochondrial disease

Hannah Steele, Aurora Gomez-Duran, Angela Pyle, Sila Hopton, Jane Newman, Renae J. Stefanetti, Sarah J. Charman, Jehill D Parikh, Langping He, Carlo Viscomi, Djordje G. Jakovljevic, Kieren G Hollingsworth, Alan J. Robinson, Robert W. Taylor, Leonardo Bottolo, Rita Horvath, Patrick F. Chinnery

Review timeline:

Submission date:	10 October 2019
Editorial Decision:	12 November 2019
Revision received:	4 December 2019
Editorial Decision:	7 January 2020
Revision received:	29 January 2020
Accepted:	29 January 2020

Editor: Céline Carret

Transaction Report:

1st Editorial Decision

12 November 2019

Thank you for the submission of your manuscript to EMBO Molecular Medicine. We have now heard back from the three referees whom we asked to evaluate your manuscript.

You will see from the comments pasted below that all three referees are supportive of publication and all mention the importance of the study. Still, explanations, clarifications, rewriting have to be done and / or expanded. Further, figure presentation is not always optimal and ref. #2 requests additional data, statistical analyses and better referencing to support the findings.

We would therefore welcome the submission of a revised version within three months for further consideration and would like to encourage you to address all the criticisms raised as suggested to improve conclusiveness and clarity. Please note that EMBO Molecular Medicine strongly supports a single round of revision and that, as acceptance or rejection of the manuscript will depend on another round of review, your responses should be as complete as possible.

Please also contact us as soon as possible if similar work is published elsewhere. If other work is published, we may not be able to extend the revision period beyond three months.

I look forward to receiving your revised manuscript.

***** Reviewer's comments *****

Referee #1 (Comments on Novelty/Model System for Author):

Bezafibrate has for some time been proposed as a potential drug for the treatment of mitochondrial diseases but has not been tested in a scientific manner in human subjects. I think it is important to publish these results

Referee #1 (Remarks for Author):

This manuscript describes for the first time a scientific evaluation of the effect of Bezafibrate in humans with mitochondrial disease. It is important to convey these results to the "mitochondrial diseases community "

Some amendments are still warranted in the manuscript. Introduction; please expand and shortly explain the difference between mtDNA vs nuclear DNA encoded diseases. Mention and add reference/s to review/s including ongoing trials with other substances. Mention that Bezafibrate has been tested not only in animal models but also in cells from mito patients.

Safety; some of the results seem to be in the safety section - consider moving these to the results section (acylcarnitines). Discussion; Is there any report showing that different elevations of FGF21 and GDF 15 is really correlated to clinical severity? Mention that this study deals with a specific subgroup of mtDNA mutation and the result might not apply for other mito diseases

Figures 1F 2G are a bit difficult to follow the individual patients. Could dots be connected?

Minor comment; "we saw" could be replaced by "we detected/observed"

Referee #2 (Comments on Novelty/Model System for Author):

This is an exploratory human clinical trial, no models used

Referee #2 (Remarks for Author):

This is an important and carefully designed early clinical study to test the safety and early efficacy of bezafibrate x12 weeks in 6 human adult patients with variable heteroplasmy for mtDNA m.3243A>G (tRNA-Leu) pathogenic variant. They authors performed a remarkably extensive series of analyses, which seemed to show that while acylcarnitines were reduced to suggest target effect, no substantial clinical benefit was seen on mitochondrial myopathy (exercise tolerance, myopathy, NMDAS score) although significant improvement in cardiac function (ES score) was seen. Extensive laboratory studies in muscle pre/post therapy and blood/urine were performed, which showed no improvement in mitochondrial oxphos function or mass, which is important since this therapy is supposed to improve mitochondrial biogenesis. Further, key "biomarkers" of mitochondrial dysfunction were exacerbated rather than improved on therapy (GDF-15, FGF21), and metabolite and RNAseq profiles suggested OXPHOS function may be compromised on therapy.

While this work is important and should be published, the manuscript as written could be improved:

1. It is not clear what the standard dosing of bezafibrate is for its approved indication (dyslipidemia), and how the dosing used in this study (600-1200 mg/day) compared -- A supp Table describes comparative dosing to mouse, but nowhere in the text is it clear whether patients were similarly/over/under dosed compared to dyslipidemia patients, all of which could impact interpretation of results. The discussion first paragraph suggests doses used were "higher than typically used", but it is not clear to what degree or discussed if it is known if going higher on dose leads to off-target effect and reduced efficacy in dyslipidemia. Further, it is not clear why no assessment of plasma lipoprotein profiles was performed, since mitochondrial disease patients often have dyslipidemia, and this is the key target effect of bezafibrate.

2. It is not clear what "evidence of myopathy as determined by the investigator" in E1 table was used. In Table 1, a baseline NMDAS score is given (although no score range or description of severity of myopathy based on this score, which would be important for the general reader), but no change over time in this score or any metric of muscle function.

3. There are quite a few typos and grammatical errors throughout that should be carefully reviewed and corrected. Similarly, the figures should be carefully reviewed to make sure all axes are labeled,

and units provided (ie, Fig 1F, what units were the ETC activities reported in?). mtDNA mutations should be written with the same nomenclature in all instances (m.3243A>G, not mt.3243A>G, etc). Abbreviations should be defined in text or legends in first use ("W0", "W12", P1, etc

4. Fig 1A is very unclear and not very helpful as written -- what does "analysis" or "markers" mean as written in multiple places -- this figure if an overview of the study could be much more informative of all the analyses that actually were performed, and clinical assessments should be included as well to give a proper overview of the study for the reader.

5. The manuscript often doesn't provide statistical results to indicate whether a result is/not significance, such as Fig 1B with FGF21 and GDF15 (neither does it indicate this in the legend).

6. The results section and discussion on metabolomic changes is incomplete and confusing, especially regarding which tissues metabolomics profiling data is being assessed in and the full significance of the altered analytes or pathways. It seems a few pathways were cherry-picked to show/discuss, without real clarity of their significance. The meaning of altered histidine metabolism is unclear (which is shown in Fig 1C to have "KEGG pathway impact" but not clear if this is increased or decreased). Glutamine and glutamate seem elevated, which standardly means there was RBC lysis during sample preparation, but this possibility was not discussed here. The presentation of the metabolomic volcano plots is confusing, with labels seemingly randomly displayed (and not just on the same side of those increased or decreased in the sample).

7. The conclusion of the manuscript should be narrowed to m.3243>G disease, not all "Mitochondrial disease".

8. What were BMIs of subjects before/during/after study? It says no weight loss occurred, but did lean body mass or total BMI change?

9. The clinical meaning of marginally improved muscle quadrupole immunofluorescence without any change in western blots of the same readouts or clinical endpoints is not clear.

10. The conclusion ends stating that future studies should "proceed with caution, with careful monitoring for possible toxicity". However, what exactly are the possible toxicity markers that should be monitored, since no toxicity was seen except for unspecified impact alterations in unbiased metabolomic screen?

11. The data suggest there is a benefit on cardiac function (end systolic volume and index) in m.3243A>G patients., yet this is not discussed in the Discussion. Was the level of change that was statistically significant also clinically meaningful? Is this a known benefit of bezafibrate, and should Bezafibrate be used to target cardiac function in m.3243A>G disease?

12. The first paragraph of the discussion suggests that dose-dependent increase in serum GDF15 and FGF21 "raises the possibility that treatment led to an exacerbation of mitochondrial pathology". However, isn't it also possible that these are simply not very specific markers of mitochondrial pathology, as shown by others, rather indicating bezafibrate effects on muscle or other organs independent of mitochondrial function?

13. Introduction section: should use primary reference for mito disease incidence (not a review article). Also, there have been pre-clinical mouse studies showing the negative impact of bezafibrate in some models that should be cited to show a more balanced perspective on its potential utility in human disease.

Referee #3 (Comments on Novelty/Model System for Author):

Human subjects with the m.3243A>G pathogenic variant were studied. As this is a common mtDNA mutation, this cohort is important to study.

Referee #3 (Remarks for Author):

Dr. Steele and colleagues have performed open-label pilot study of bezafibrate, a PPAR-alpha agonist, in 6 subjects with mitochondrial myopathy due to the m.3243A>G mitochondrial DNA (mtDNA) mutation. Participants received bezafibrate 600mg daily for 6 weeks followed by 1200mg daily for 6 additional weeks. This study is important because preclinical experiments have indicated that this drug may be therapeutic for mitochondrial myopathies by enhancing mitochondrial biogenesis. Safety, clinical outcome measures, as well as blood, urine, and muscle biomarkers, were analyzed. No clinically significant adverse events were observed and reductions in cytochrome c oxidase (COX) deficient fibers suggested benefit. However, increases in FGF-21 and GDF-15 as well as amino acid and tricarboxylic acid (TCA) cycle intermediates were interpreted as indicators of possible progression of the mitochondrial myopathy. The study not designed to detect clinical changes. Alterations in the biochemical and metabolic profiles appear robust, indicate target engagement, and warrant publication. Minor comments are listed below.

1. Discussion, page 5, par 1: Change phrase to "...observed dose-dependent increases in GDF-15 and FGF-21"
2. Discussion, page 5, par 2: The reference for the sentence "In the context of mtDNA mutations..." should be Chen, Kirk et al., 2018 because the 2 cited references describe other cell models.
3. In the methods section, the authors should note the muscle biopsy technique (needle vs. open biopsy) that was presumably conducted with ethical committee approved consent forms.

1st Revision - authors' response

4 December 2019

Referee #1

Referee: Introduction; please expand and shortly explain the difference between mtDNA vs nuclear DNA encoded diseases.

Response: We have expanded the introductory paragraph as follows:

"Mitochondrial diseases are genetically determined metabolic disorders of oxidative phosphorylation (OXPHOS) caused by mutations of mitochondrial DNA (mtDNA) or nuclear genes encoding mitochondrial proteins."

Referee: Mention and add reference/s to review/s including ongoing trials with other substances. Mention that Bezafibrate has been tested not only in animal models but also in cells from mito patients.

Response: We have added the following text to the introduction:

"Several new strategies are being developed as possible treatments for mitochondrial disorders. These include protein replacement; small molecules including antioxidants, amino acids and nucleotide supplementation; cell therapy; and gene manipulation (Nightingale, Pfeffer et al., 2016). Although pre-clinical studies have seen the greatest progress, several are now under evaluation in clinical trials (Steele, Horvath et al., 2017)."

and

"Bezafibrate has the most extensive pre-clinical evidence of efficacy in animal models and patient cell lines (Hofer, Noe et al., 2014)."

Referee: Safety; some of the results seem to be in the safety section - consider moving these to the results section (acylcarnitines).

Response: We have moved the acylcarnitine section to the "Biochemistry and genetics" section of the Results.

Referee: Discussion; Is there any report showing that different elevations of FGF21 and GDF 15 is really correlated to clinical severity? Mention that this study deals with a specific subgroup of mtDNA mutation and the result might not apply for other mito diseases.

Response: We agree with the reviewer and have deleted reference to the severity of mitochondrial disease in the context of FGF21 and GDF15. We have also made it clear that our results refer to a specific genetic subgroup of mitochondrial disease patients, including addition of the following text to the last paragraph of the discussion:

“It should be noted that our results are based on observations in a specific sub-group of patients with mitochondrial disease due to the m.3243A>G mutation.”

Referee: Figures 1F 2G are a bit difficult to follow the individual patients. Could dots be connected?

Response: We have revised the figure as suggested (note the numbering has changed with other revisions to the figures).

Referee: Minor comment; "we saw" could be replaced by "we detected/observed".

Response: We have replaced 'we saw' with the proposed alternatives throughout the manuscript.

Referee #2

Referee: It is not clear what the standard dosing of bezafibrate is for its approved indication (dyslipidemia), and how the dosing used in this study (600-1200 mg/day) compared -- A supp Table describes comparative dosing to mouse, but nowhere in the text is it clear whether patients were similarly/over/under dosed compared to dyslipidemia patients, all of which could impact interpretation of results. The discussion first paragraph suggests doses used were "higher than typically used", but it is not clear to what degree or discussed if it is known if going higher on dose leads to off-target effect and reduced efficacy in dyslipidemia.

Response: We have explained this by including the following text in the 'Participant and Treatment' section:

“The standard dose of bezafibrate used to treat dyslipidemia in the UK is 200mg three times daily (TDS).”

Referee: Further, it is not clear why no assessment of plasma lipoprotein profiles was performed, since mitochondrial disease patients often have dyslipidemia and this is the key target effect of bezafibrate.

Response: Non-fasting triglyceride levels were measured before and after treatment. There was a trend towards a reduction, but mean level across all of the participants did not change. We have added the following to the Results section and included the results in a new Table EV5:

“Although there was a trend towards a reduction in non-fasting triglyceride levels after 12-weeks of treatment, the mean level across all participants did not change significantly (Table EV5).”

Referee: It is not clear what "evidence of myopathy as determined by the investigator" in E1 table was used. In Table 1, a baseline NMDAS score is given (although no score range or description of severity of myopathy based on this score, which would be important for the general reader), but no change over time in this score or any metric of muscle function.

Response: We have clarified how we defined the presence of myopathy in the text as follows, and we have included the NMDAS data in a new supplementary table (Table EV11). The revised text includes the following:

“All had clinical evidence of a myopathy on bedside testing (Medical Research Council grade 4+ or less).”

and

“The mean questionnaire scores did not change following treatment, including the NMDAS and muscle-specific scores (Table E11).”

Referee: There are quite a few typos and grammatical errors throughout that should be carefully reviewed and corrected. Similarly, the figures should be carefully reviewed to make sure all axes are labelled and units provided (ie, Fig. 1F, what units were the ETC activities reported in?). mtDNA mutations should be written with the same nomenclature in all instances (m.3243A>G, not mt.3243A>G, etc). Abbreviations should be defined in text or legends in first use ("W0", "W12", "P1", etc.)

Response: We thank the reviewer for pointing these errors out. We have carefully proof-read the manuscript, added new labels to Fig. 1F (now Fig. 2E) including the units of measurement, corrected all nomenclature to m.3243A>G, and defined the abbreviations in the text of each figure legend.

Referee: Fig. 1A is very unclear and not very helpful as written -- what does "analysis" or "markers" mean as written in multiple places -- this figure if an overview of the study could be much more informative of all the analyses that actually were performed, and clinical assessments should be included as well to give a proper overview of the study for the reader.

Response: We agree with the reviewer and have completely redesigned this panel, now shown as Fig. 1.

Referee: The manuscript often doesn't provide statistical results to indicate whether a result is/not significant, such as Fig. 1B with FGF21 and GDF15 (neither does it indicate this in the legend).

Response: We have added the statistical significance to the figures (as symbols), which are also explained in the text of the figure legend.

Referee: The results section and discussion on metabolomic changes is incomplete and confusing, especially regarding which tissues metabolomics profiling data is being assessed in and the full significance of the altered analytes or pathways. It seems a few pathways were cherry-picked to show/discuss, without real clarity of their significance. The meaning of altered histidine metabolism is unclear (which is shown in Fig. 1C to have "KEGG pathway impact" but not clear if this is increased or decreased). Glutamine and glutamate seem elevated, which standardly means there was RBC lysis during sample preparation, but this possibility was not discussed here. The presentation of the metabolomic volcano plots is confusing, with labels seemingly randomly displayed (and not just on the same side of those increased or decreased in the sample).

Response: We have redesigned all of the volcano plots in the main figures and Supplementary Material. Following the reviewer's suggestion, in Fig. 2B and Supplementary Fig. EV4, the labels are displayed separately for decreasing (left) and increasing (right) metabolites and ordered vertically according to the level of significance. For ease of interpretation, we also colour-coded the metabolite labels according to the identified KEGG pathways (dark red = "Alanine, aspartate and glutamate metabolism", dark green = "Histidine metabolism" and dark blue = "Citrate cycle (TCA cycle)", with a transition colour-code if the metabolite belongs to more than one KEGG pathways). Apart from "Citrate cycle (TCA cycle)", where metabolites that are all decreasing (Fig. 2B and Supplementary Fig. EV4C&D), in both "Alanine, aspartate and glutamate metabolism" and "Histidine metabolism" pathways, metabolites are both increasing and decreasing.

As discussed in the Supplementary Material and shown in Table EV7, significantly enriched pathways are selected based on a conservative 10% adjusted Holm–Bonferroni (H-B) threshold. The next KEGG pathway identified in the analysis ("Pyrimidine metabolism") has both a level of significance (probability of rejecting the null hypothesis) after H-B correction and an "Impact" that is three times smaller than "Citrate cycle (TCA cycle)". For clarity, we have now highlighted the adjusted H-B *P*-value for the selected KEGG pathways in Fig 2C.

When comparing metabolite changes between 0 and 6&12-week treatment, "L-Glutamic acid" and, to a less extent, "L-Glutamine" are elevated. However, they are not significantly different when comparing metabolite changes between 6 and 12 weeks (Fig. EV4B) and only "L-Glutamic acid" shows a significant increase after 6-weeks of treatment (Fig. EV4C), suggesting that glutamine and glutamate elevated values are not due to systematic artefacts in sample preparation.

Finally, we have expanded our discussion to include further details on the histidine, with relevant additional citations, as follows:

Similar findings have been observed for aspartate, with effects on reductive carboxylation (Chen et al., 2018, Sullivan, Luengo et al., 2018). Likewise increased histidine has been observed in models of mitochondrial disease (Chen et al., 2018, Vo, Paul Lee et al., 2007), where it can act a source of folate (Mao, Vyas et al., 2004) and TCA cycle pre-intermediates such as glutamate, which are then degraded to α -ketoglutarate (Alifano, Fani et al., 1996).

Referee: The conclusion of the manuscript should be narrowed to m.3243>G disease, not all "Mitochondrial disease".

Response: We have revised the last paragraph to make it explicit that our results are based on observations in one specific genetic subgroup of patients with mitochondrial disease.

Referee: What were BMIs of subjects before/during/after study? It says no weight loss occurred, but did lean body mass or total BMI change?

Response: There was no change in BMI. We have included the data in a new table and comment on this in the results section as follows:

“There was no change in mean body mass index (BMI) during the treatment (Table EV4).”

Referee: The clinical meaning of marginally improved muscle quadrupole immunofluorescence without any change in western blots of the same readouts or clinical endpoints is not clear.

Response: We agree with the reviewer, and have added this text to the discussion to make this clear:

“Thus, although the quadruple immunofluorescence may be a sensitive biomarker indicative of a treatment response, the clinical significance of these findings remains to be determined.”

Referee: The conclusion ends stating that future studies should "proceed with caution, with careful monitoring for possible toxicity". However, what exactly are the possible toxicity markers that should be monitored, since no toxicity was seen except for unspecified impact alterations in unbiased metabolomic screen?

Response: We agree with the reviewer, and have added more specific guidance at the end of the discussion:

“Measuring blood metabolomic biomarkers, FGF-21, and GDF-15 in addition to serum creatine kinase would provide some reassurance if levels do not change on treatment.”

Referee: The data suggest there is a benefit on cardiac function (end-systolic volume and index) in m.3243A>G patients., yet this is not discussed in the Discussion. Was the level of change that was statistically significant also clinically meaningful? Is this a known benefit of bezafibrate, and should Bezafibrate be used to target cardiac function in m.3243A>G disease?

Response: We agree with the reviewer, and have added the following section to the discussion:

“It is encouraging that we observed an improvement in parameters of cardiac function on MRI that are known to be impaired in patients with m.3243A>G (Hollingsworth, Gorman et al., 2012). The clinical significance of these findings is unclear at present, but they support the use of cardiac MRI as a biomarker to monitor treatment in mitochondrial diseases (Steele et al., 2017), which is sensitive to changes over a short period and correlates with the underlying biochemical defect in skeletal muscle (Hollingsworth et al., 2012).”

Referee: The first paragraph of the discussion suggests that dose-dependent increase in serum GDF15 and FGF21 "raises the possibility that treatment led to an exacerbation of mitochondrial pathology". However, isn't it also possible that these are simply not very specific markers of

mitochondrial pathology, as shown by others, rather indicating bezafibrate effects on muscle or other organs independent of mitochondrial function?

Response: We agree with the reviewer and have added this caveat to the opening paragraph of the Discussion.

“Although given the lack of specificity of GDF-15 and FGF-21, it is also plausible that the changes we observed were due to the direct effects of bezafibrate on metabolism independent of oxidative phosphorylation.”

Referee: “Introduction section: should use primary reference for mito disease incidence (not a review article). Also, there have been pre-clinical mouse studies showing the negative impact of bezafibrate in some models that should be cited to show a more balanced perspective on its potential utility in human disease.”

Response: We have added a primary reference to the epidemiology of mitochondrial disease (Gorman et al 2015) to the introduction. We have also added the following text explaining the side effects of bezafibrate noted in rodents.

“Bezafibrate has a favourable side-effect profile in humans, but has been noted to cause liver toxicity in rodents, including mouse models of mitochondrial disease (Dillon et al., 2012, Viscomi et al., 2011, Yatsuga & Suomalainen, 2012a).”

Referee #3

Referee: Discussion, page 5, par 1: Change phrase to "...observed dose-dependent increases in GDF-15 and FGF-21"

Response: We have corrected the typo highlighted by the reviewer.

Referee: Discussion, page 5, par 2: The reference for the sentence "In the context of mtDNA mutations..." should be Chen, Kirk et al., 2018 because the 2 cited references describe other cell models.

Response: We have corrected the reference as suggested.

Referee: In the methods section, the authors should note the muscle biopsy technique (needle vs. open biopsy) that was presumably conducted with ethical committee approved consent forms.

Response: We have added the following text to the methods section:

“Needle muscle biopsies were performed with ethical approval and patient consent.”

2nd Editorial Decision

7 January 2020

Thank you for the submission of your revised manuscript to EMBO Molecular Medicine. We have now received the enclosed report from the referee who was asked to re-assess it. As you will see, the reviewer is now supportive, and I am pleased to inform you that we will be able to accept your manuscript pending the following final amendments:

1) Please address the text changes commented by referee 2 and provide additional details and clarifications as needed.

Please submit your revised manuscript within two weeks. I look forward to seeing a revised form of your manuscript as soon as possible.

***** Reviewer's comments *****

Referee #2 (Comments on Novelty/Model System for Author):

This reports an open-label study in 6 m.3243A>G adult patients with myopathy. It is the first human study, although extensive preclinical studies have previously been reported in the literature.

Referee #2 (Remarks for Author):

Overall, the authors have made substantially improvements in the manuscript quality and clarity and been responsive to most reviewer feedback review. However, several points remain unclear and require further revision:

ABSTRACT:

1. Given the corrections and clarifications in the manuscript, the abstract as written appears to be misleading/overstated in several places:

A. The abstract should be modified to lessen the strength of the conclusion that indirect GDF15/FGF21/metabolomics markers "are signs of compromised oxidative phosphorylation". These are "soft signs" at best, and not direct indicators of oxphos activity.

B. Similarly, the last statement states bezafibrate "worsened the metabolomic signature of mitochondrial disease". However, while exacerbations of non-specific abnormalities were seen, it's not clear these were "worse" or negative, as implied, since it's not known if these metabolic changes are adaptive or maladaptive in the setting of m.3243A>G disease and therapeutic response.

C. Further, given the modest cardiac findings (as discussed in Discussion) and the inconsistent muscle complex IV results that were not present on western blot, this statement is overstated/misleading and should be more precisely qualified: "a reduction in the number of complex IV immunodeficient muscle fibres and improved cardiac function."

D. The authors end the abstract and Discussion cautioning about safety and including a recommendation to measure FGF21, GDF15, and CK.

a. Was CK measured in this study, and did it change with treatment?

b. In the general dyslipidemic patients (or original clinical trials that gained bezafibrate approval), was muscle dysfunction seen, and even if these markers go up, is that definitely a sign of toxicity? Otherwise, this warning seems overstated in this current context and not highly compelling.

RESULTS:

1. Participants and Treatments, Paragraph 1. "All had clinical evidence of myopathy on bedside testing (Medical Research Council grade 4+ or less)." Can the authors clarify if "myopathy" is used to mean "weakness", and proximal/distal, as opposed to fatigability (which is more characteristic of "mitochondrial myopathy"?). Also, there seems to be a missing % in front of female in the written text: "50-84% heteroplasmy, female, ..." Finally, were the 6 subjects unrelated?

2. It is surprising and concerning that lipid levels were not significantly reduced with bezafibrate for 12 weeks in this population (Table EV5), and questions whether the doses used were adequate for basic target engagement in this population, let alone desired muscle effects. This should be included in first 2 sentences of discussion regarding whether/not they had target engagement in their study.

3. Safety and Tolerability section of results. It seems misleading to say there is "enrichment" of TCA metabolites, when Fig 2B and EV4 shows all of the TCA metabolites are decreased (not increased) with treatment

a. Given this finding, it is confusing how the authors interpreted analyte changes in the Discussion: "Glutamine is an abundant circulating amino acid and has an anaplerotic role, replenishing TCA cycle intermediates which drive the mitochondrial respiratory chain to generate reducing equivalents (Chen, Kirk et al., 2018, Mullen, Hu et al., 2014). In the context of mtDNA mutations, glutamine is reduced by the TCA cycle leading to the synthesis of aspartate which is essential for nucleotide synthesis (Chen et al., 2018). Thus, increased amino acid levels, including glutamate, could be part

of a compensatory response to the primary OXPHOS defect (Buzkova et al., 2018, Chen et al., 2018)."

i. Specifically, if increased glutamine (which is less than 2-fold here) were to increase TCA intermediates, this is not what is shown in the data here, where all TCA intermediates are decreased. Please review and further clarify or revise this section of the Discussion.

4. Table 1 and Table EV11 - NMDAS score values/range for overall or subparts still isn't clarified anywhere in table legends or text to assist in interpretation of the severity indicated by the numbers shown. Is a low or high score considered good or bad, what is max for each part, what is the CMID that would make a change with treatment meaningful or significant, etc. The general reader will not know how to intuitively interpret these raw data as shown without context.

5. Fig 2B - metabolomic volcano plots are now easier to interpret. However, it's not clear what the identity is of the 4-fold upregulated cluster of metabolites and single downregulated metabolite. These would see important, and perhaps more telling of what impact bezafibrate is having then focusing only on pathway identity of several of the 2-3-fold modified metabolites? Since metabolism is the stated focus of this trial, these findings should be clarified and discussed.

6. Fig 2C 2D - The analogous question as for Fig 2B applies to clarifying what are the most dysregulated acylcarnitine species (Fig 2D) and what is the clearly outlying orange circle pathway in Fig 2C (with KEGG pathway impact at 0.7)?

7. Fig 3B - even though there is marginal sig at level of $p < 0.05$ for Complex IV immunofluorescence, the data themselves aren't very impressive in terms of magnitude of pre-treatment elevation or response to change. It appears one individual (P6) had most response, with perhaps modest response in P1 and no response in other 4 subjects. Should these considerations soften the interpretation of these data by the authors in abstract and discussion?

8. Fig 3E - it's still not clear in figure or legend if these "altered" pathways are upregulated or downregulated or treatment.

9. Fig 3G is perhaps the most important outcome of this open-label study. However, it's not clear what the "normal" value or range would be for these metrics, and in the text the authors don't state if baseline values are abnormal in these 6 m.3243A>G subjects? This is highly interesting and important and should be clarified to help readers interpret the meaning of the significant improvement in these metrics identified in these individuals.

10. Table EV5 - the units of the triglyceride measures shown are not indicated, nor the normal range. Were the pre-treatment levels elevated in any or across the mean of all 6 subjects?

11. Supplemental Methods files explicitly describes MRS was done in skeletal muscle and heart, but the heart MRS data don't appear to be shown or discussed in Fig1/results/discussion section.

Supplemental Tables and Figures Files:

The numbering and labelling of many of these files are very confusing as labelled in the electronic journal review system when looking at the individual PDFs (excel files appear different in terms of headings but the overall tables are different than as labelled in system and referred to in text):

Table EV6 - when open PDF, is labelled "Table E4. Differential metabolites analysis. Light red fill with dark red text cells indicate metabolites that ar", which has nothing to do with the content of the actual table.

Table EV7 - when open PDF, is labelled "Table E5. Pathway enrichment analysis of significant differential metabolites. Ligh values are obtained by Benjamini and Hochberg (1995) method, while impact va each pathway."

Table EV9 - when open PDF, is labelled "Table E7. Differential RNA-seq analysis. Light red fill with dark red text cells indicate genes that are si". This label has nothing to do with the actual content of the table.

Table EV10 - when open PDF is labelled "E8"

Also, in Urea/Creatine/GFR data figures, the individual subjects don't appear to be labelled anywhere. It's not clear what figure # this is, perhaps fig EV1?

2nd Revision - authors' response

29 January 2020

Response to Referee #2

Referee #2: The abstract should be modified to lessen the strength of the conclusion that indirect GDF15/FGF21/metabolomics markers "are signs of compromised oxidative phosphorylation". These are "soft signs" at best, and not direct indicators of oxfhos activity.

Response: We have deleted "are signs of compromised oxidative phosphorylation".

Referee #2: Similarly, the last statement states bezafibrate "worsened the metabolomic signature of mitochondrial disease". However, while exacerbations of non-specific abnormalities were seen, it's not clear these were "worse" or negative, as implied, since it's not known if these metabolic changes are adaptive or maladaptive in the setting of m.3243A>G disease and therapeutic response. Further, given the modest cardiac findings (as discussed in Discussion) and the inconsistent muscle complex IV results that were not present on western blot, this statement is overstated/misleading and should be more precisely qualified: "a reduction in the number of complex IV immunodeficient muscle fibres and improved cardiac function."

Response: We have changed 'worsened' to 'altered'

Referee #2: The authors end the abstract and Discussion cautioning about safety and including a recommendation to measure FGF21, GDF15, and CK. Was CK measured in this study, and did it change with treatment?

Response: CK was measured in this study. There was no change in the mean level before and after treatment.

Referee #2: In the general dyslipidemic patients (or original clinical trials that gained bezafibrate approval), was muscle dysfunction seen, and even if these markers go up, is that definitely a sign of toxicity? Otherwise, this warning seems overstated in this current context and not highly compelling.

Response: We agree with the reviewer. Our findings highlight the need for caution, as opposed to demonstrating that bezafibrate is contra-indicated in this patient group. The manuscript is worded accordingly.

Referee #2: Participants and Treatments, Paragraph 1. "All had clinical evidence of myopathy on bedside testing (Medical Research Council grade 4+ or less)." Can the authors clarify if "myopathy" is used to mean "weakness", and proximal/distal, as opposed to fatiguability (which is more characteristic of "mitochondrial myopathy"?). Also, there seems to be a missing % in front of female in the written text: "50-84% heteroplasmy, female, ..." Finally, were the 6 subjects unrelated?

Response: All of the patients had proximal muscle weakness. We have revised the text accordingly. We thank the reviewer for highlighting the missing text (the genders were only specified in the table). 4 of the participants were female. We have added this to the text. Finally, the participants were not related. We have indicated this in the revised text.

Referee #2: It is surprising and concerning that lipid levels were not significantly reduced with

bezafibrate for 12 weeks in this population (Table EV5), and questions whether the doses used were adequate for basic target engagement in this population, let alone desired muscle effects. This should be included in first 2 sentences of discussion regarding whether/not they had target engagement in their study.

Response: Although there was not a statistically significant change in the mean level of triglycerides before and after treatment ($P=0.08$), there was a trend for them to decrease (shown in Appendix Table S5). We have drawn attention to this at the start of the Discussion.

Referee #2: Safety and Tolerability section of results. It seems misleading to say there is "enrichment" of TCA metabolites, when Fig 2B and EV4 shows all of the TCA metabolites are decreased (not increased) with treatment

Response: We agree that 'enrichment' is misleading. This word was chosen because the analytical approach is called 'enrichment analysis. To avoid any confusion, we have replaced the word 'enrichment' with 'highlighted changes in'.

Referee #2: Given this finding, it is confusing how the authors interpreted analyte changes in the Discussion: "Glutamine is an abundant circulating amino acid and has an anaplerotic role, replenishing TCA cycle intermediates which drive the mitochondrial respiratory chain to generate reducing equivalents (Chen, Kirk et al., 2018, Mullen, Hu et al., 2014). In the context of mtDNA mutations, glutamine is reduced by the TCA cycle leading to the synthesis of aspartate which is essential for nucleotide synthesis (Chen et al., 2018). Thus, increased amino acid levels, including glutamate, could be part of a compensatory response to the primary OXPHOS defect (Buzkova et al., 2018, Chen et al., 2018)." Specifically, if increased glutamine (which is less than 2-fold here) were to increase TCA intermediates, this is not what is shown in the data here, where all TCA intermediates are decreased. Please review and further clarify or revise this section of the Discussion.

Response: We agree with the reviewer, and have added the following section to the discussion to address this point:

However, given the observed increase in all three amino acids, it remains puzzling that, overall, the TCA metabolites decreased on treatment. The reasons for this are not clear, but could reflect the effects of increased aspartate levels which are known to inhibit alpha-ketoglutarate utilisation (Chen et al., 2018), with knock-on effects on other TCA cycle intermediates.

Referee #2: Table 1 and Table EV11 - NMDAS score values/range for overall or subparts still isn't clarified anywhere in table legends or text to assist in interpretation of the severity indicated by the numbers shown. Is a low or high score considered good or bad, what is max for each part, what is the CMID that would make a change with treatment meaningful or significant, etc. The general reader will not know how to intuitively interpret these raw data as shown without context.

Response: We have added the maximum score and indicated the direction of the severity in the legend of the table. The legend includes the following text:

Each section contains a number of sub-sections which are scored from 0 to 5, with 5 being the most severe. Section 1 includes 10 sub-sections (maximum score 50), Section 2 includes 9 sub-sections (maximum score – 45), Section 3 includes 10 sub-sections (maximum score – 50), giving an overall maximum score of 145 in the most severely affected patients. The sub-scores for the muscle-specific sections are shown in the lower table.

Referee #2: Fig 2B - metabolomic volcano plots are now easier to interpret. However, it's not clear what the identity is of the 4-fold upregulated cluster of metabolites and single downregulated metabolite. These would see important, and perhaps more telling of what impact bezafibrate is having then focusing only on pathway identity of several of the 2-3-fold modified metabolites? Since metabolism is the stated focus of this trial, these findings should be clarified and discussed.

Response: All of the results are presented in the additional data tables (now Datasets EV1-4), allowing the reader to directly address any specific questions along the lines of those highlighted by

the reviewer. We are reluctant to focus our discussion on the larger magnitude changes, because it is known that the size of the change does not necessarily correspond to the importance of the change in terms of its effect on metabolism.

Referee #2: 6. Fig 2C 2D - The analogous question as for Fig 2B applies to clarifying what are the most dysregulated acylcarnitine species (Fig 2D) and what is the clearly outlying orange circle pathway in Fig 2C (with KEGG pathway impact at 0.7)?

Response: Again, the reader can enquire directly using the published datasets EV1-4. The orange symbol in Fig 2C is the "Synthesis and degradation of ketone bodies" KEGG pathway. Only one significantly different metabolite ("acetoacetate") was identified in this pathway (out of six) with a small pathway enrichment *P*-value (*y*-axis).

Referee #2: Fig 3B - even though there is marginal sig at level of $p < 0.05$ for Complex IV immunofluorescence, the data themselves aren't very impressive in terms of magnitude of pre-treatment elevation or response to change. It appears one individual (P6) had most response, with perhaps modest response in P1 and no response in other 4 subjects. Should these considerations soften the interpretation of these data by the authors in abstract and discussion?

Response: We agree with the reviewer that there is variability in terms of the response. We drew attention to this in the Discussion section. We are very reluctant to discuss individual cases and have restricted our discussion to the statistically significant results across the group. Focussing on individuals could be considered 'cherry picking' and is known to be potentially misleading.

Referee #2: Fig 3E - it's still not clear in figure or legend if these "altered" pathways are upregulated or downregulated or treatment.

Response: We have presented the data in the standard format used in pathway analysis. The idea of the figure is to show whether the pathways are disrupted or not. The reader can look at the individual metabolites (shown in the published datasets) to see whether or not any one individual metabolite is up, or down-regulated.

Referee #2: Fig 3G is perhaps the most important outcome of this open-label study. However, it's not clear what the "normal" value or range would be for these metrics, and in the text the authors don't state if baseline values are abnormal in these 6 m.3243A>G subjects? This is highly interesting and important and should be clarified to help readers interpret the meaning of the significant improvement in these metrics identified in these individuals.

Response: We have cited a reference which includes normal data and data from m.3243A>G patients.

Referee #2: Table EV5 - the units of the triglyceride measures shown are not indicated, nor the normal range. Were the pre-treatment levels elevated in any or across the mean of all 6 subjects?

Response: We have included the normal reference range and units in the table legend. Lipid levels were significantly increased in 2 individuals

Referee #2: Supplemental Methods files explicitly describes MRS was done in skeletal muscle and heart, but the heart MRS data don't appear to be shown or discussed in Fig1/results/discussion section.

Response: We have deleted the additional unnecessary methods.

Referee #2: Supplemental Tables and Figures Files: The numbering and labelling of many of these files are very confusing as labelled in the electronic journal review system when looking at the individual PDFs (excel files appear different in terms of headings but the overall table #s are different than as labelled in system and referred to in text):

Response: We have updated the naming and numbering of the tables and datasets in accordance with the Editors recommendations.

Corresponding Author Name: Patrick Chinnery, Rita Horvath, Leo Bottolo

Manuscript Number: 2019-11589